# The Role of Deubiquitinating Enzyme in Head and Neck Squamous Cell Carcinoma

**DOI:** 10.3390/ijms24010552

**Published:** 2022-12-29

**Authors:** Shengjian Jin, Yasusei Kudo, Taigo Horiguchi

**Affiliations:** Department of Oral Bioscience, Tokushima University Graduate School of Biomedical Sciences, Tokushima 770-8504, Japan

**Keywords:** head and neck squamous cell carcinoma, deubiquitination, ubiquitination

## Abstract

Ubiquitination and deubiquitination are two popular ways for the post-translational modification of proteins. These two modifications affect intracellular localization, stability, and function of target proteins. The process of deubiquitination is involved in histone modification, cell cycle regulation, cell differentiation, apoptosis, endocytosis, autophagy, and DNA repair after damage. Moreover, it is involved in the processes of carcinogenesis and cancer development. In this review, we discuss these issues in understanding deubiquitinating enzyme (DUB) function in head and neck squamous cell carcinoma (HNSCC), and their potential therapeutic strategies for HNSCC patients are also discussed.

## 1. Ubiquitin System

Ubiquitin is a highly-conserved small regulatory protein that has been found in almost all tissues of eukaryotic organisms [1]. Ubiquitin was first identified in 1975. It performs its myriad functions through conjugation to a large range of target proteins. A variety of different modifications can occur. This discovery that ubiquitin can be attached to proteins and label them for destruction won the Nobel Prize for chemistry in 2004 [2]. The ubiquitin protein consists of 76 amino acids and has a molecular mass of about 8.5 kDa. Under the conditions where ATP provides energy, ubiquitin molecules bind to the target protein through the cascade catalytic reaction of ubiquitin-activating enzyme (E1), ubiquitin-conjugating enzyme (E2), and ubiquitin ligase (E3). The ubiquitinated target protein is recognized and degraded by 26S proteasome. At present, 2 E1s, about 50 E2s, and more than 600 E3s are known to be encoded by the human genome. The multiplexed complex cascade catalytic networks reflect the importance and complexity of the ubiquitin system [3].

The C-terminal Gly residues of ubiquitin bind to the ε-amino groups of the target protein Lys residues via labeling amide bonds to the target protein. The ubiquitin itself also contains seven Lys residues: K6, K11, K27, K29, K33, K48, and K63. Ubiquitin can form polyubiquitin chains by connecting with their own Lys. Polymer chains are formed through a variety of connections between substrates and ubiquitin molecules. The chain can be short, containing only one ubiquitin molecule (mono-ubiquitin), or multiple sites on the substrate which are connected to a ubiquitin molecule (polyubiquitin), or it can be long. If it extends on the same Lys residue of a ubiquitin molecule, it is called a homogeneous ubiquitin chain (for example, K48- or a K63-linked polyubiquitin chain). If the ubiquitin molecule is connected with different Lys residues, the chain has a mixed topology, which is called a mixed ubiquitin chain [4]. All types of ubiquitin modifications can be detected in cells [5,6], and different types of ubiquitin modifications may lead to different results in cells, indicating that ubiquitin can be used as a code to store and transmit information. In general, polyubiquitin-labeled proteins that are linked by K48 are recognized and degraded by 26S proteasome. Additionally, the polyubiquitin of K63 connections affects the function of proteins more and participates in the process of endocytosis, DNA damage repair, activation of protein kinase, transformation of protein subcellular localization, some signal transduction, and stress response [7,8,9]. The polyubiquitination of the K6 connection can be involved in the regulation of microtubule stability and mitotic spindle orientation [10]. Polyubiquitin that is linked by K27 can participate in DNA damage repair and autoimmune regulation [11]. The polyubiquitination of K6 and K33 connections can be involved in DNA damage repair [12]. Polyubiquitin that is linked by K33 is also involved in the Type I interferon signaling pathway and innate immunity and vesicular transport [13]. The same type of ubiquitination may produce different cellular functions, and different types of ubiquitination may cooperate with each other to play a role in the same process or function, or one ubiquitination of a protein will promote the occurrence of another ubiquitination. Thus, the target protein gives a specific biological function via ubiquitination. The complex relationship between ubiquitin types and cell function leads to the abnormality of ubiquitin which may lead to complex diseases and tumors (Figure 1).

## 2. Deubiquitinating Enzymes (DUBs)

DUBs maintain ubiquitin system homeostasis by cleaving polyubiquitin chains or completely removing ubiquitin chains from ubiquitinated proteins. Via deubiquitination, free ubiquitins are generated and recycled [14]. Deubiquitination has important functions in regulating the ubiquitin-dependent pathways including cell cycle regulation, cell death, protein degradation, protein function, gene expression, and signal transduction [15]. So far, approximately 100 DUBs have been identified and classified as eight different families: ubiquitin-specific peptidases (UBP/USP) family, ubiquitin C-terminal hydroxylases (UCH) family, JAB1/MPN/MOV34 proteases (JAMM) family, ovarian tumor proteases (OTU) family, Machado–Joseph domain (MJD) family, motif interacting with Ub-containing novel DUB family (MINDY), monocyte chemotactic protein-induced proteins (MCPIP) family, and the Zn-finger and UFSP domain protein (ZUFSP) family [16,17] (Table 1). Among them, the USP, UCH, OTU, MJD, MCPIP, MINDY, and ZUFSP families belong to cysteine hydrolases, while the JAMM family belongs to metalloenzymes. The protein hydrolysis method that is adopted by most DUBs is to separate the ε-amino group of the target protein Lys residue from the carboxyl group of the C-terminal Gly of the ubiquitin molecule.

The USP family is the largest DUB family, which is called UBP family in yeast. The USP/UBP family is species- and tissue-specific, and the species and quantity vary across different organisms or different tissues. Members of the USP family contain three highly conserved domains: finger, palm, and thumb domains. The active catalytic center is in the palm and thumb domains, while the finger domain is used to capture the Gly residues at the end of ubiquitin molecules. Some USP domain activities are strongly inhibited by the slow dissociation of ubiquitin after substrate hydrolysis [18].

The UCH family was first discovered and reported as a member of the family of mercapto proteases, which mainly modify the ubiquitin chain. Its catalytic core region contains 230 amino acids. In the apo-enzyme structure of ubiquitin C-terminal hydrolase isozyme L3 (UCHL3), a prominent loop covers the active site. Surprisingly, in the ubiquitin aldehyde-bound structure, this active-site crossover loop straddles the C-terminal residues of the bound ubiquitin. Such an arrangement may prevent the binding of large and folded ubiquitin conjugates by fitting the substrates through this confined loop. Even polyubiquitin chains would be too big to be threaded through this confined loop [19]. Due to this unique spatial structure of UCH, it has a certain selectivity to the substrate. Therefore, members of the UCH family are only involved in the degradation of small peptides, such as those that are produced by proteasome and lysosomal pathways, or can also degrade protein precursors in the process of extension.

The JAMM/MPN family belongs to the metalloproteinase family. Members of the family usually couple two zinc ions, one in the active center and the other in the ubiquitin recognition region. The JAMM domain can recognize the tripeptide sequence Gln62-Lys63-Glu64. Therefore, the JAMM family can specifically recognize the ubiquitin chain that is connected by K63 [20]. The JAMM family is the only DUB that binds to ubiquitin molecules at the active site of the ubiquitin enzyme [21]. The JAMM family is not sensitive to DUB inhibitors but can be inhibited by metastatic cationic chelating agents.

The OTU family is divided into three subgroups: OTU, OTUB, and A20-like OTU. The members of the OTU family are not homologous to the USP and UCH families, but they have different specificities for different types of ubiquitin chains. For example, ubiquitin aldehyde binding 1 (OTUB1) can only hydrolyze the K48 ubiquitin chain. Ubiquitin aldehyde binding 2 (OTUB2) can hydrolyze the K63 and K48 ubiquitin chain. A20 can only hydrolyze K48 ubiquitin chain. Zinc finger RANBP2-type containing 1 (TRABID, also known as ZRANB1) can recognize the ubiquitin chain of K29 and K33 [22,23]. The activity conversion mode of the OTU family is similar to that of the USP family. When there is no ubiquitin binding, it is inactivated. When the ubiquitin molecule binds, it exposes the catalytic active site and activates enzyme activity [24].

The most widely studied member of the MJD family is ATXN3, a protein that is associated with degenerative diseases. ATXN3 has two ubiquitin binding sites, one of which is located at the active site and the other group is far away from the active site. Therefore, ATXN3 can hydrolyze the ubiquitin labeling of the substrate by interacting with the distal ubiquitin binding site to form a stable open concept. It recognizes the ubiquitin chains that are connected by K48 and K63, but it is more specific to K63 [25].

MCPIP is a newly discovered DUBs family. The first member of this family is zinc finger CCCH-type containing 12A (MCPIP1), At the N terminus, MCPIP1 contains a ubiquitin association domain (UBA). UBA mediates the association of MCPIP1 with ubiquitinated proteins, but it does not affect the enzymatic activity. In the middle, there is a conserved CCCH-type Zn finger domain (ZF), which is commonly responsible for its catalytic activity, and MCPIP1 contains a Pro-rich domain at the C terminus. There is no essential sequence similarity between MCPIP1 and the five DUB domains except remote homology with the UCH domain (27%), suggesting that MCPIP1 may contain a novel DUB domain [26].

MINDY and ZUFSP were discovered in 2016 and 2018, respectively [17,27,28]. Both have no homology with any known other DUB families. The MINDY family is highly selective at cleaving K48 ubiquitin chain [29]. MINDY may have specialized roles in regulating proteostasis, but ZUFSP is highly selective at cleaving the K63 ubiquitin chain and plays an important role in preventing DNA damage [30].

## 3. The Relationship between DUB and Tumorigenesis

Various DUBs have been reported to have connections to tumor-suppressing or oncogenic functions and may, therefore, represent potential therapeutic targets. The tumor suppressor p53 plays a critical role to preserve DNA fidelity from diverse insults through the regulation of cell-cycle checkpoints, DNA repair, senescence, and apoptosis [31]. MDM2 proto-oncogene (MDM2), an E3 ubiquitin ligase, is the main regulator of the p53 protein [32]. It can induce p53 to enucleate and degrade. Additionally, a cytoplasmic DUB, ubiquitin-specific peptidase 10 (USP10) deubiquitinates p53 and prevents Mdm2-induced p53 nuclear export and degradation. After DNA damage, USP10 is stabilized and translocated to the nucleus to activate p53 [33]. When subjected to external pressure, ubiquitin-specific peptidase 42 (USP42) can bind to p53 and stabilize its protein by removing the ubiquitin chain. Thus, the quick activation of the p53 signaling pathway causes the cell cycle arrest and carries out DNA repair for preventing carcinogenesis [34]. Ubiquitin-specific peptidase 15 (USP15) can stabilize MDM2, as well as degrade transcription factor NFATc2, to regulate T-cell activation. The deletion of USP15 activates T-cells and enhances their response to bacterial infection and tumorigenesis [35]. OTU deubiquitinase 1 (OTUD1) is a member of the OTU domain DUB family, which can directly inhibit the ubiquitination of p53 that is mediated by MDM2. It can stabilize and activate p53, resulting in p53-dependent cell proliferation inhibition and apoptosis. DNA damage enhances the interaction between OTUD1 and p53, so OTUD1 primarily plays a role in the G2/M and S checkpoints of the cell cycle [36]. OTUD1 inhibits DNA repair by inhibiting the activity of Ubc13/RNF168. This control of ring finger protein 168 (RNF168) affects the ubiquitination of chromatin. When DNA is damaged, OTUD1 is temporarily dissociated from the Ubc13/RNF168 complex, causing RNF168 to catalyze the ubiquitination of chromatin at DNA damage sites. At this site, it is associated with the UbcH5/MDM2 complex, which inhibits MDM2-dependent ubiquitination of p53, and p53 becomes stable and activated [37]. OTU deubiquitinase 5 (OTUD5) can also deubiquitinate p53 protein and stabilize it, thus affecting apoptosis that is induced by DNA damage [38].

Ubiquitin-specific peptidase 4 (USP4) is an important p53 regulatory molecule, which directly binds to ARF-BP1 and removes ubiquitin, resulting in the stabilization of the ARF-BP1 complex and subsequent decrease of p53 levels, and inhibits p53-related apoptosis. The absence of USP4 is thought to promote senescence. USP4 is highly expressed in various cancers, including bladder, prostate, and thyroid, indicating that it is a potential oncogene [39]. Ubiquitin-specific peptidase 7 (USP7) is also a p53 regulatory factor, which is able to deubiquitinate and stabilize p53 [22].

Transcription factor KLF transcription factor 5 (KLF5) is a member of the BAP1/HCF-1 complex, which is expressed in breast cancer tissues and promotes cancer cell proliferation and metastasis. BRCA1-associated protein 1 (BAP1) stabilizes KLF5 by removing the ubiquitin chain, thereby promoting the occurrence and metastasis of breast cancer. BAP1 depletion can inhibit the growth and metastasis of hepatocellular carcinoma cells [40].

Cell division cycle 25A (Cdc25A) is a phosphatase that promotes cell cycle progression by activating cyclin-dependent kinase (CDK), which has the potential to promote cancer. Cdc25A is highly expressed in tumor tissues and can be stabilized by ubiquitin-specific peptidase 17 like family member 2 (Dub3). Overexpression of Dub3 induces cell cycle arrest at S and G2 phases and activates the DNA damage response via stabilization of Cdc25A protein. Higher expression of Dub3 promotes the high levels of Cdc25A in breast cancer [41].

Phosphatase and tensin homolog (PTEN) is a tumor suppressor gene, which is deleted in many tumors. Ubiquitin-specific peptidase 13 (USP13) can deubiquitinate the PTEN protein and stabilize it. In breast cancer cells without USP13, the PTEN protein was downregulated, while Akt protein phosphorylation, cell proliferation, soft agar colony formation, glycolysis, and tumor growth were significantly enhanced [42,43]. OTU deubiquitinase 3 (OTUD3) stabilizes the PTEN protein via cleaving its ubiquitin chain. OTUD3 depletion activates the Akt signaling pathway, cell transformation, and metastasis. Reduced expression of OTUD3 is involved in carcinogenesis of breast cancer via the downregulation of PTEN protein [34,42,44,45].

Ubiquitin-specific peptidase 22 (USP22) participates in histone deubiquitination and acetylation, thus affecting gene transcription. It affects tumorigenesis and metastasis through its effects on BMI-1, MYC, FBP1, and TRF1.

Thus, there are many different functions of DUBs, which directly affect the stability of many kinds of proteins in cells and are closely related to the occurrence and development of tumors. Currently, however, the research on DUBs is not deep enough, and the function of many DUBs has not been confirmed. By comprehensively understanding the relationship between DUBs and tumorigenesis, screening DUBs that are closely related to tumor progression, and thoroughly studying the action mechanism of DUBs in tumors, the targeted therapy of DUBs can be really applied to clinical anti-tumor therapy.

## 4. The Role and Mechanism of DUB in HNSCC

HNSCC was the seventh most common type of cancer worldwide in 2018 [46], representing about 6% of all cases and accounting for an estimated 650,000 new cancer cases and 350,000 cancer deaths worldwide every year [47]. The pathogenic factors include tobacco and alcohol intake, human papillomavirus (HPV) or Epstein Barr virus (EBV) infection, radiation, periodontal disease, vitamin deficiency, and eating habits [48]. Achieving a better understanding of molecular aberrations that are associated with HNSCC might identify new diagnostic and therapeutic strategies for this disease. (Table 2, Figure 2) [49,50,51,52,53,54,55,56,57,58,59,60,61,62].

## 5. The Role of DUBs in HNSCC Cell Proliferation and Apoptosis

Cell cycle regulation, uncontrolled proliferation, and inhibition of apoptosis will affect the occurrence and development of tumors. USP22, which has been described as a cancer stem cell (CSC) marker [68,69], is a member of the largest subfamily of DUBs [70]. The USP22 gene is located on chromosome 17 and consists of 14 exons, can indirectly affect chromatin structure through histone ubiquitination (H2A and H2B), thus regulating the transcriptional activation of many genes and widely affecting biological functions [66]. USP22 has the protein characteristics of CSCs and can promote the formation of various proteins for invasive tumor growth [71]. In addition, USP22 has been shown to regulate cellular growth and proliferation. Indeed, USP22 depletion reduces transformation of c-Myc and plays an important role in cell cycle progression and anchorage-independent growth [72]. Additionally, the downregulation of USP22 results in tumor cells remaining arrested at the G1 phase via p21 stabilization [73,74]. Thus, USP22 plays an important role in cell cycle regulation. The depletion of USP22 in HNSCC tissue can affect the CDK inhibitor (CDKI)/Rb signal pathway. USP22 increases the expression of cyclin-dependent kinase inhibitor 1A (CDKI) including p21 and p27, and it decreases the expression of Rb protein in HNSCC cells [74,75,76]. An immunohistochemical study showed that USP22 expression is upregulated in laryngeal carcinoma and has a close relationship with malignant behaviors including invasion, metastasis, and poor prognosis [77].

Ubiquitin-specific protease 9X (USP9X) is related to the occurrence and development of pancreatic cancer, multiple myeloma, and other tumors [78]. USP9X knockdown in HNSCC cells decreases the percentage of cells in the G0/G1 phase and increases the percentage of cells in the S and G2/M phase. The proliferation of HNSCC cells can be regulated by downstream transcription factor HES-1 (Notch pathway) through USP9X level [65]. USP9X may also regulate the proliferation of HNSCC through the mammalian target of the rapamycin (mTOR) pathway [77,79]. However, the loss of USP9X may be harmful to the growth of tumor cells, but its loss in primary tumor cells may accelerate the development of secondary tumors [80].

USP14 has been proven to be related to the occurrence and development of many cancers. In tumor xenograft mice, the growth of HNSCC with low USP14 was slower than that of the control group. The expression of USP14 in tongue carcinoma was higher than that in adjacent tissues. USP14 depletion can inhibit the proliferation and migration of HNSCC cells in vitro [81]. Moreover, USP14 can promote the proliferation and invasion of HNSCC both in vivo and in vitro.

Ubiquitin-specific peptidase 4 (USP4) has been identified to deubiquitinate K63-linked ubiquitin conjugates from TNF receptor-associated factor 2 (TRAF2), TNF receptor associated factor 6 (TRAF6), and TGF-beta activated kinase 1 (TAK1) and stabilizes molecules by deubiquitinating K48-linked ubiquitination [63]. USP4 is significantly upregulated in tumor tissues compared to matched non-tumor tissues in HNSCC. Co-immunoprecipitation showed that USP4 interacted with receptor-interacting protein 1 (RIPI) to remove K63-connected ubiquitin molecules to stabilize RIPI expression. Thus, it negatively regulates the activation of NF-κB mediated by deubiquitination of RIPI. These findings indicate that USP4 has tumor suppressor roles in HNSCC [63].

## 6. Relationship between DUBs and Prognosis of HNSCC

The prognosis of HNSCC is related to its stage of recurrence and lymph node metastasis. USP7 participates in intracellular tumor regulation, DNA repair, immune response, and epigenetic effects [82]. An immunohistochemical study showed that USP7 overexpression is correlated with histone-lysine N-methyltransferase (EZH2) in HNSCC [61]. EZH2 can promote the epithelial-mesenchymal transformation (EMT) in vivo and inhibits cellular senescence and differentiation [83]. EZH2 is widely expressed in various malignant tumors, including nasopharyngeal carcinoma [84]. USP7 expression in poorly differentiated HNSCC is higher than that in well differentiated HNSCC, and the high expression of USP7 was significantly related to a high degree of lymphatic invasion and high TNM stage. Survival analysis showed that USP7 overexpression is correlated with poor prognosis in HNSCC patients [61].

USP2a is one of the two splicing variants of USP2, which regulates the stability and function of many important cell growth and differentiation regulators and signal transduction factors in vivo [85]. In HNSCC, increased USP2a is associated with poor prognosis. The expression of ErbB2-interacting protein (Erbin) and USP2a increased in HNSCC, indicating that the expression of Erbin and USP2a was positively correlated with lymph node metastasis. Moreover, USP2a expression is related to epidermal growth factor (EGF) in HNSCC [86].

## 7. The Role of DUBs in Radiosensitivity of HNSCC

For the patients with HNSCC in the advanced stage, surgery combined with radiotherapy is needed to improve the survival rate of the patients. The radiation target and dose should be determined according to the location of the primary tumor, clinical-stage, postoperative pathological diagnosis, and postoperative imaging evaluation. The tolerance of HNSCC to radiotherapy is the main factor affecting the prognosis. BRCA1-associated protein-1 (BAP1), also known as UCHL2 can regulate various cellular processes including cell cycle, cell differentiation, transcription, DNA damage response, and resistance to tumor radiotherapy. BAP1 mutation can increase the sensitivity of HNSCC to radiotherapy and lead to tumor susceptibility syndrome, which is not related to the status of high-risk human papillomavirus (HPV) and p53 tumor suppressor protein. It was found that the survival fraction of BAP1 knockout HNSCC was significantly lower than that of HNSCC cells after radiotherapy, the sensitivity to radiotherapy was increased, and this sensitivity could be reversed by forced re-expression [67]. BAP1 can enhance the stability of histone H2A by deubiquitination and increase its expression level, thus interfering with the modification of chromatin and histone at the double strand break, reducing the rate of cell proliferation and increasing the sensitivity of radiotherapy [87]. HNSCC that is driven by HPV is more sensitive to DNA-damaging therapy than HPV-negative HNSCC. A recent study revealed that p16, the clinically used surrogate for HPV positivity, renders cells more sensitive to radiotherapy via the p16–HUWE1–USP7–TRIP12 pathway [88].

## 8. The Role of DUBs in Immune Mechanisms of HNSCC

Cancer is considered to be an immunosuppressive disease that is characterized by dysfunction of immune cells and disturbance of cytokine excretion [89]. EBV is closely related to the occurrence of HNSCC [90]. BPLF1 is the largest protein in EBV. There is deubiquitinating activity in the first 205 amino acids of EBV proteins. Its deubiquitinating activity can cleave K63 and K48 polyubiquitin chains. These two regulatory functions play a role in preventing virus protein degradation [91]. When the BPLF1 of EBV was knocked out, When the BPLF1 of EBV was knocked out, the replication quantity and infectivity of its genome decreased. The BPLF1 protein of EBV and its conserved deubiquitinating activity regulate cellular DNA repair enzyme polymerase and recruit it to potential virus destruction and replication sites, thus enhancing the ability to produce infectious viruses. BPLF1 can increase the fault tolerance rate of EBV virus after infecting human DNA by deubiquitinating and stabilizing EBV virus polymerase Pol, so that the DNA of the virus can continue to replicate in the injured site, thus increasing the infectivity of the virus [92]. It has been found that BPLF1 inhibits NF-κB signal transduction during lysozyme infection via deubiquitinating TNF receptor-related factor 6 (TRAF6) [93]. As HNSCC is closely related to EBV infection, the existence of BPLF1 may promote the HNSCC development.

USP9X upregulation can lead to programmed death receptor-ligand 1 (PD-L1) deubiquitination and PD-L1 stable expression in HNSCC tissues. PD-L1 inhibits T-cell activation and proliferation and allows cancer cells to escape T-cell immune surveillance. Preventing USP9X from locating PD-L1 may be an effective strategy for the treatment of HNSCC, especially metastatic tumors [64].

The conserved cylindromatosis (CYLD) belongs to the family of USPs, and its deletion can activate NF-κB and MAPK signaling pathways in HNSCC. NF-κB transcription factors can promote cell survival and induce innate and adaptive immunity to pathogens, such as viruses or bacterial pathogens, inflammatory cytokines, which are abnormally activated during the occurrence, and the development of cancer. The main target of CYLD is the classical NF-κB signal transduction that is mediated by NF-κB essential regulatory protein (NEMO) and cytokine reactive IκB kinase (IKK) complex subunit IKK-γ. CYLD deubiquitinates NEMO and prevents the phosphorylation of IκB and NF-κB activation [94]. The loss of CYLD in HNSCC can also increase the invasiveness of HNSCC by promoting transforming growth factor beta receptor 1 (ALK5) to stabilize NF-κB.

## 9. The Therapeutic Effect of DUBs on HNSCC

As DUBs have become a clinical target for tumor treatment, many people are paying more and more attention to the application of DUBs targeting drugs in cancer therapy. As most ubiquitin enzymes are cysteinases, it is easier to develop into a candidate drug. The effects of DUB inhibitors are as follows: (1) to increase the aggregation of polyubiquitin molecules, (2) to reduce the mono-ubiquitin part, and (3) to change some molecular activities, especially the persistence of carcinogenic proteins that are mediated by DUBs [16]. The inhibition of DUBs can lead to functional impairment of the proteasome and the accumulation of misfolded proteins, resulting in tumor cytotoxicity and death [95]. Some DUBs are associated with cancer and can be used as targets for new inhibitors [96,97]. At present, by screening the DUB inhibitor library, it was found that compound 2X-0324 had a potential radio-sensitizing effect on laryngeal cancer (RER > 1–1.86). Vif1 and Vif2 are protease inhibitors of USP7, which can regulate the anticancer effect that is mediated by p53 and have the potential to become molecular targeted drugs for the treatment of laryngeal carcinoma [98]. Azepan-4-ones and b-AP15 are protease inhibitors of USP14 and UCHL5. They can induce the accumulation of high molecular weight polyubiquitin proteins, resulting in the enhancement of the unfolded protein response (UPR). Both can inhibit the development of HNSCC [99]. In tongue cancer, b-AP15 can significantly inhibit the growth of tumor cells and increase apoptosis in a dose-dependent manner, thereby overcoming the drug resistance of bortezomib in vitro [100].

Squamous cell carcinoma of the head and neck is the most common malignant tumor of the head and neck, and its occurrence and development are related to a variety of DUBdeubiquitination enzymes [49,50,51,52,53,54,55,56,57,58,59,60,61,62,63,64,65,66,67]. Some DUBs are not only functional in head and neck squamous cell carcinomas but are also important in other squamous cell carcinomas. For example, CYLD is associated with HPV infection and inhibits NF-κB signaling in HNSCC, further promoting the growth and metastasis of head and neck cancer. The same functions of CYLD and HPV occur in cervical cancer cells [101]. In addition, USP46 is also associated with HPV infection in esophageal squamous cell carcinoma, but it has not been reported in HNSCC [102]. The Hippo/YAP signaling pathway plays an important role in HNSCC [103]. USP36 promotes proliferation and invasion of esophageal squamous cell carcinoma by deubiquitination of YAP [104]. USP36 may also play an important role in head and neck cancer.

## 10. Conclusions and Future Perspectives

In this review, we summarized the current information on the structure, regulation, and function of DUBs in tumor development, especially the association and treatment of DUBs and HNSCC. The following keywords were used to find the literature and the year in this paper: “deubiquitinating enzyme” and “head and neck cancer”. The literature was searched from 1997 to 2022 by PubMed.

DUBs can reverse the ubiquitination of the target protein, maintain the balance between ubiquitination and deubiquitination of the substrate protein, and maintain cell homeostasis by reducing the amount of deubiquitination in the target protein. DUBs have four different mechanisms: processing ubiquitin protein precursors, recovering ubiquitin molecules in the ubiquitin process, cutting ubiquitin protein chains, and reversing the binding of ubiquitin proteins to substrates [95]. DUBs regulate many cellular functions in vivo, including proteasome-dependent protein degradation [95], gene expression [105], cell cycle progression [106], chromosome segregation [107], DNA repair [108], and apoptosis [109]. The increased expression of USP22 and USP7 is related to the poor clinical prognosis of HNSCC and can be used as a biomarker to evaluate the prognosis of HNSCC. The expression of BAP1 can increase the radiotherapy tolerance in response to HNSCC and is a potential target to improve the radiosensitivity of HNSCC. The expression of USP9X, USP14, and other DUBs is increased in HNSCC, and it can promote the proliferation of cancer cells. The silencing of USP4 can inhibit the apoptosis of cancer cells and increase the persistent survival rate of HNSCC, as immunosuppressive diseases, UXP9X, CYLD1, and BPLF1 accelerate the development of HNSCC through the escape inhibition mechanism of immune cells or promoting the spread of virus infection. DUB is gradually being regarded as a therapeutic target for cancer because of its structural advantages. DUB inhibitors such as 2X-0324 and b-AP15 may serve as a new therapeutic strategy for HNSCC.

Our understanding of DUBs function, mechanism of action, regulation, and disease associations has made tremendous progress over the past decade. But DUBs-focused drug discovery has been challenging because the mechanism of action of DUB enzymes is often complex. The functions of DUBs are concentrated on the substrates that interact with DUBs. The same DUBs can have several or even dozens of substrates, and different DUBs can also have the same substrate, and the function of the substrates may interact with each other, which makes the design of predictive biochemical analysis and the development of drug-like compounds challenging. Although we can predict the existence of a wide range of therapeutic potentials for DUBs, a great deal of research is needed to successfully screen and evolve small molecule DUBs inhibitors in the future.

At present, a variety of literature has reported that DUB plays an important role in the occurrence and development of HNSCC, including abnormal expression in tumor cells and close correlation with radiotherapy and immune mechanism. Although the mechanism and downstream effects that are regulated by DUBs are not clear, as a potential target for the treatment of HNSCC, it is particularly important to better study the site of this regulatory mechanism and the side effects of targeted therapy. With the deepening of the study on the mechanism and function of DUBs in HNSCC, it is possible for DUBs to be used as a molecular target in cancer treatment.

## Figures and Tables

**Figure 1 ijms-24-00552-f001:**
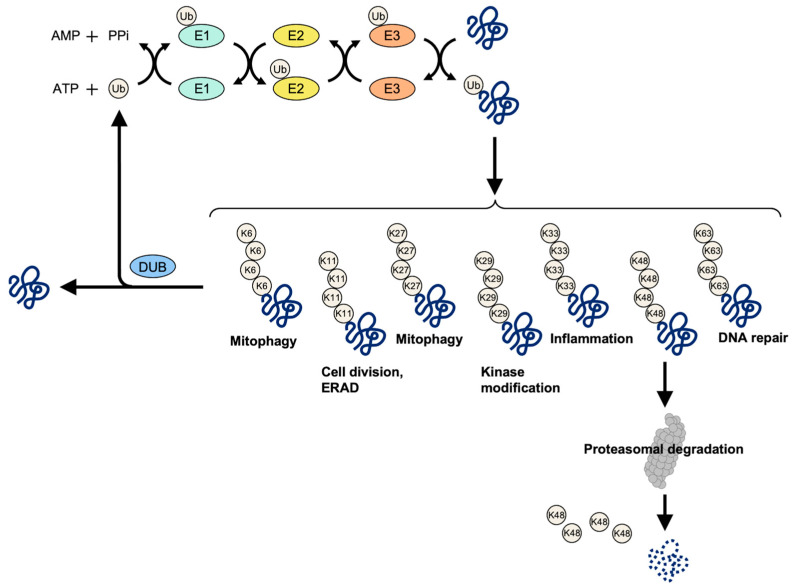
Schematic of key events in ubiquitination and deubiquitination. Under the conditions where ATP provides energy, ubiquitin binds to the target protein through the cascade catalytic reaction of ubiquitin-activating enzyme (E1), ubiquitin-conjugating enzyme (E2), and ubiquitin ligase (E3). The ubiquitinated target protein is recognized and degraded by 26S proteasome. At present, 2 E1s, about 50 E2s, and more than 600 E3s are known to be encoded by the human genome. Deubiquitinating enzymes (DUBs) maintain ubiquitin system homeostasis by cleaving polyubiquitin chains or completely removing ubiquitin chains from ubiquitinated proteins. Via deubiquitination, free ubiquitins are generated and recycled.

**Figure 2 ijms-24-00552-f002:**
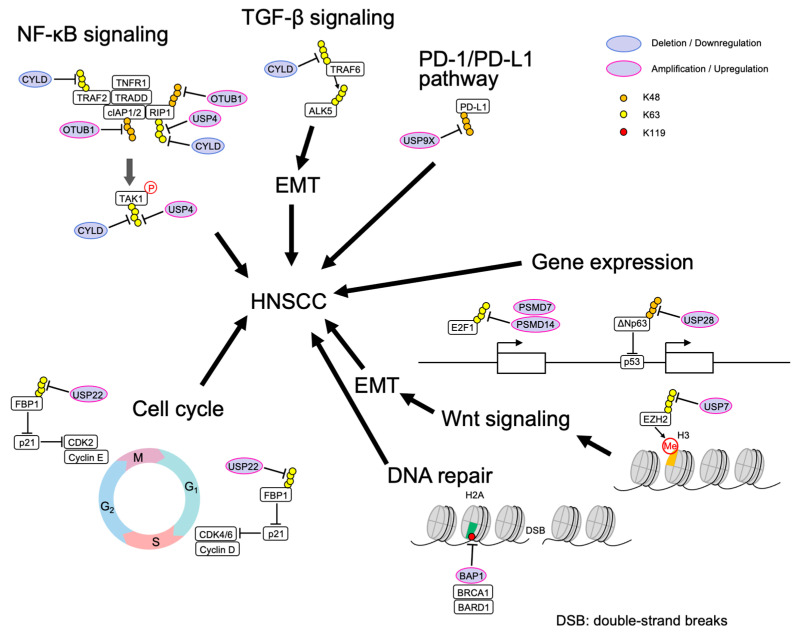
Involvement of DUBs in HNSCC.

**Table 1 ijms-24-00552-t001:** DUB families and their members.

DUB Family		Type	Member
USPs	ubiquitin-specific proteases	cysteine protease	USP1-USP8, USP9X, USP9Y, USP10-USP16, USP18-USP22, USP24-USP26, USP27X, USP28-USP54, CYLD, DUB3
UCHs	ubiquitin carboxy-terminal hydrolases	cysteine protease	BAP1, UCHL1, UCHL3, UCHL5
JAMMs	JAB1/MPN/MOV34 proteases	zinc-dependent metalloprotease	BRCC3, COPS5, PSMD, MPND, MYSM1, STAMBP, STAMBPL1, PRPF8, COPS6, EIF3F, PSMD7, EIF3H
OTUs	otubain/ovarian tumor-domain containing proteins	cysteine protease	OTUB1, OTUB2, OTUD1, OTUD3, OTUD4, OTUD5, OTUD6A, OTUD6B, OTUD7A, OTUD7B, OTULIN, OTULINL, TNFAIP3, ZRANB1, YOD1, VCPIP1
MJDs (Josephins)	Machado–Joseph disease domain superfamily	cysteine protease	ATXN3, ATXN3L, JOSD1, JOSD2, JOSD3
MCPIPs	monocyte chemotactic protein- induced protein	cysteine protease	MCPIP1, MCPIP2, MCPIP3, MCPIP4, MCPIP5, MCPIP6, MCPIP7
MINDYs	motif interacting with ubiquitin- containing novel DUB family	cysteine protease	FAM63A/MINDY1, FAM63B/MINDY2, FAM188A/MINDY3, FAM188B/MINDY4
ZUFSP/SUP1	Zn-finger and UFSP domain protein	cysteine protease	ZUP1

**Table 2 ijms-24-00552-t002:** DUB function in HNSCC.

DUB	Abnormal Regulation	The Roles in HNSCC	Substrates	References
CYLD	mutation	removes K63-polyubiquitin and M1 linear-ubiquitin chains and inhibits NFκB signaling	RIP1, TRAF2, TRAF6, TAK1, NEMO	[49]
	Low expression	promotes TGF-β signaling and cell invasion	ALK5	[50]
	Low expression	removes K63-polyubiquitin, thereby inhibiting TGF-β signaling	SMAD7	[51,60]
USP4	overexpression	removes K63-polyubiquitin and promotes TNF-α induced apoptosis	RIP1	[63]
USP7	overexpression	promotes cell growth, cell migration, and invasion	EZH2	[52,61]
USP9X	overexpression	deubiquitinates and stabilizes PD-L1 to promote cell proliferation	PD-L1	[64]
	Low expression	mTOR pathway		[65]
USP22	overexpression	associates with lymph node metastasis and histological grade		[66]
USP28	overexpression	inhibits p53 on the promoter of pro-apoptotic genes	Np63	[53,54,55,56]
BAP1	overexpression	deubiquitinates H2A at the DSB site suppresses transcription, and promotes DNA repair	H2Aub(K119)	[67]
PSMD14	overexpression	stabilizes E2F1, gives stemness to cells by SOX2 expression and Akt signal activation	E2F1	[57]
PSMD7	overexpression	a prognostic factor correlated with immune infiltration		[58]
OTUB1	overexpression	a risk factor		[49,59,62]

## Data Availability

No new data were created or analyzed in this study.

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
