# Peer review of "The Role of Deubiquitinating Enzyme in Head and Neck Squamous Cell Carcinoma"

_ijms, 2022, doi:10.3390/ijms24010552_

Round 1

Reviewer 1 Report

- Chapter 2 should be renamed as 'Deubiquitinating enzymes' as it is not really the deubiquitination system.

- Table 1 - all DUBs should be included, not sufficient to include only a few USP members as all DUBs in other families seem to be include. Space can be saved by writing as USP1-17 etc.

- It its Epstein Barr virus (EBV), not EB virus. It should also be clarified that EBV is associated with Nasopharyngeal carcinoma, which is not HNSCC

- The parapraph 'BA-125, an inhibitor of USP14, can cause a large increase 239 of ubiquitin protein, trigger tumor cell apoptosis, and activate caspase-3 and PARP-1 240 cleavage, thus causing tumor cell apoptosis [79].' should be removed. The inhibitor is b-AP15, not BA-125, and reference 79 is discussing DLBCL, not HNSCC

- A recent study on USP7 driving radiation resistance in HNSCC should also be cited (Molkentine et al., Nat Comms, 2022)

- Comparisons to the role of DUBs in other squamous cell carcinomas would be of great benefit in the manuscript (Cervical cancer - CYLD, An et al, Cancer cell, 2006; USP46, Kiran et al., Mol Cell, 2018; USP13, Morgan et al., Oncogene, 2021. ESCC - USP36, Zhang et al., Cell Death and Disease, 2022. 

- Ref 84 is not for BAP1 as stated in Table 2. Please check all references thoroughly

Reviewer 2 Report

Shengjian Jin et al., have discussed the role of deubiquitination in HNSCC pathogenesis.  In my opinion, the manuscript can be accepted with some minor modifications/changes.

1. The authors should add one section or paragraph; what keywords do they use to find the literatus and the year? 

2. I can only see one figure in the whole manuscript; they should add at least one more figure for better clarity.

3. The authors have used lots of Acronyms for enzymes/genes; they should mention the full name at the place of first-time use.

4. Conclusion section is a bit small; they can also add limitations and future aspects of the study.

5.  Since deubiquitination and ubiquitination are normal processes for the cell, how are the authors sure that this can be used for HNSCC targeting? Kindly mention it in the manuscript at an appropriate place. 
